# Use of Fecal Indices as a Non-Invasive Tool for Ruminal Activity Evaluation in Extensive Grazing Sheep

**DOI:** 10.3390/ani12080974

**Published:** 2022-04-09

**Authors:** Carla Orellana, Giorgio Castellaro, Juan Escanilla, Víctor H. Parraguez

**Affiliations:** 1Doctoral Program in Forest, Agricultural and Veterinary Sciences, Faculty of Agricultural Sciences, University of Chile, Santiago 8820808, Chile; carla.orellanam@gmail.com; 2Department of Animal Production, Faculty of Agricultural Sciences, University of Chile, Santiago 8820808, Chile; juanescanillacruzat@gmail.com (J.E.); vparragu@uchile.cl (V.H.P.); 3Department of Animal Biological Sciences, Faculty of Veterinary Sciences, University of Chile, Santiago 8820808, Chile

**Keywords:** annual Mediterranean climate-type rangeland, extensive grazing evaluation, fecal indices, sheep nutrition, rumen activity, volatile fatty acids

## Abstract

**Simple Summary:**

The main objective of this work was to evaluate the degree of association that exists between three fecal indices (concentrations of 2,6 diaminopimelic acid, nitrogen, and phosphorus) and biomarkers of ruminal activity, as a non-invasive way to estimate the nutritional status in sheep grazing on annual rangeland. It was possible to establish that fecal indices, and especially fecal nitrogen and phosphorus, were linearly and positively correlated with the ruminal concentration of some volatile fatty acids, especially branched-chain, and rumen ammonia. Due to the above, these fecal indices could be used to evaluate the ruminal activity and the nutritional status of grazing sheep, with minimal manipulation of the animals.

**Abstract:**

For a simple, non-invasive evaluation of nutritional status of sheep kept under extensive grazing conditions on annual rangeland, fecal indices (2,6 diaminopimelic acid, nitrogen, and phosphorus) obtained during the vegetative, reproductive, and dry grassland phenological stages, were correlated with ruminal physiological biomarkers (volatile fatty acids and ruminal ammonia). Through correlation analysis and linear regressions, the degree of association between the variables studied was established. The fecal indices that presented the highest degree of association with ruminal variables were FN and FP, being highly correlated with the production of branched-chain volatile fatty acids (isobutyrate and isovalerate) and with ruminal ammonia (*r* ≥ 0.65), establishing simple linear regression equations of high significance (*p* ≤ 0.05). Therefore, fecal indices, especially fecal concentrations of N and P, could reflect the metabolism at the ruminal level and with it the availability of compounds for microbial growth, which would help to establish the nutritional status of sheep herds under extensive grazing conditions.

## 1. Introduction

Fecal indices have been widely used as a tool for estimating nutritional status in ruminants, given their relationship with nutrient intake [1]. These studies have been carried out mostly in wild ungulates and seek, through minimally invasive techniques, to evaluate the content of compounds such as protein [2], dry matter digestibility [3], energy intake [4], and the dietary amount of some minerals such as phosphorus (P) [5] and calcium [6]. Fecal nitrogen (FN) would correspond to an indicator of protein intake [7]; however, its use is based on the fact that this compound is rather a reflection of the amount of microbial biomass generated under conditions of high availability of nitrogen at ruminal level. Therefore, if the amount of FN is higher, so will the production of microbial protein [8]. Something similar occurs in the case of 2,6 diaminopimelic acid (DAPA), since the synthesis of this indigestible compound depends on the proliferation of Gram (-) bacteria in the rumen [9], which would benefit from high nutritional content of the diet. The higher the content of fecal DAPA (DAPAf), the greater the bacterial growth in the rumen [10]. In the case of P, its content in feces would be directly related to the intake of this mineral [11]. The function of P at the ruminal level is crucial for cellulolytic bacteria [12] since they depend on P for the cell wall degradation [13]. When the P needs of ruminal microorganisms are not covered, microbial activity is affected [14], reducing, for example, the dry matter intake [15].

We have recently studied the relationship between fecal indices (DAPA, nitrogen, and phosphorus) and their reliability for estimation of nutrient intake in sheep under extensive grazing on an annual natural grassland, throughout the year. Interestingly, we found that fecal nitrogen and phosphorus are highly correlated, adequately reflecting the nutrient composition of the prairie in its four phenological stages. Thus, these indicators turned out to be good estimators of dry matter, protein, and energy intake in grazing sheep [16].

The validity of the use of the fecal indicators would be closely associated with ruminal activity, beyond the intake of certain types of nutrients. However, few studies have looked for such a relationship. Among the compounds that are generated as a product of dietary fermentation, volatile fatty acids, or short-chain fatty acids (VFAs), and ruminal ammonia (RA) are some of the most important for ruminants [17]. The VFAs are the basis of energy metabolism; acetic, propionic, and butyric acids being the most important. The absorption of these VFAs occurs in the rumen due to a difference in concentration [18], where an increase in the production of VFAs would also increase their absorption and availability in the different metabolic pathways where they are used. An example of the above is the generation of glucose from propionic acid, and the deposition of adipose tissue in the case of acetic acid [17]. As a product of ruminal fermentation, branched-chain VFAs (isobutyric and isovaleric) are also generated, which are important since they are used by ruminal bacteria as an energy source [19]. The RA is the product of the degradation of amino acids in the rumen, so its concentration would be a reflection of the availability of nitrogen for the synthesis of proteins of microbial origin [20].

In accordance with what was mentioned in the previous paragraphs and as a continuation of our previous study, we hypothesize that fecal indices, being linearly associated with compounds derived from ruminal activity, could be used as estimators of the bacterial activity, reflecting the potential absorption of nutritionally relevant compounds for ruminants. Therefore, the objectives of this work were to establish the degree of association and mathematical equations that relate some fecal indices with the concentration of some elements derived from ruminal fermentation in sheep kept under an extensive grazing system on annual rangeland in the Mediterranean climate type in the central Chilean zone.

## 2. Materials and Methods

### 2.1. Study Area and Sheep Management

This continuity study was carried out with samples obtained from the sheep used in a previous experimental protocol; therefore, the details of the location (Figure 1) and sheep handling can be reviewed in [16]. In brief, four two-year-old non-pregnant non-lactating merino precoz ewes (average live weight of 51.6 ± 8.9 kg) were used. The animals, fitted with ruminal cannulas, were kept permanently, year-round, in a natural annual grassland (“round year” continuous grazing; 3.28 dry sheep equivalent/ha), made up mainly of a woody stratum dominated by *Vachellia caven* (Mol.) Seigler and Ebinger (“Espino”) and a herbaceous stratum composed of winter-growing therophyte species from genera *Avena*, *Aira*, *Bromus*, *Hordeum*, *Vulpia*, *Lolium*, and *Erodium* [21,22]. Sheep did not receive any type of supplement, for which the grassland plants constituted 100% of their diet (Figure 2).

### 2.2. Sheep Sampling and Measurements

In each grassland phenological stage (vegetative, reproductive, and dry), fresh fecal samples, taken directly from the rectum of each animal, were daily obtained during five consecutive days, for fecal nitrogen (FN), phosphorus (FP), and 2,6-diaminomipelic acid concentration (DAPAf) determination, as previously described by Orellana et al. [16].

In parallel to the fecal sampling, ruminal content samples for volatile fatty acids (VFAs) and ruminal ammonia (RA), were taken from the ventral, central, and caudal areas of the rumen, until obtaining approximately 400 g. Samples were subjected to a filtration using a double layer gauze and the resulting fluid was stored under different conditions, depending on the subsequent analyses [23]. The ruminal fluid samples for the determination of VFAs and RA were received in 50 mL cryovials, prepared with 1 mL of 1 N sulfuric acid, to fix the compounds of interest. These cryovials were frozen at −20 °C until their subsequent analysis. For determination of VFAs, the samples were thawed at room temperature and centrifuged at 351× *g*, at 4 °C for 10 min. The supernatant was subjected to gas chromatography to determine the concentration of acetic, propionic, butyric, valeric, isobutyric, and isovaleric acids [24]. The RA concentration was estimated through the Kaplan method [25], where the sample thawed at room temperature was centrifuged twice at 67,200× *g* for 10 min. The supernatant was exposed to a solution of hypochlorite and phenol and analyzed through spectrophotometry.

### 2.3. Statistical Analysis

Statistical processing of the results was undertaken using Statgraphics Centurion XV software. A statistical model of repeated means was used for VGAs and RA, considering grassland phenological stage (EF_i_ = vegetative, reproductive, dry) as the main fixed effect, and the animal as a nested effect within each stage (Y_ij_ = μ + EF_i_ + animal (EF)_j_ + ε_ij_). Differences among means attributed to the effect of the grassland’s phenological stages were established by using the least significant difference (LSD) test at a significance of 95%. Differences were considered significant when *p* ≤ 0.05. Data are presented as means ± standard error of mean.

Furthermore, Pearson’s or Spearman’s correlations among fecal indices obtained previously [16] and ruminal traits were calculated between variables, according to data distributions. To determine the magnitude of the association between variables, linear regressions were also calculated [26].

## 3. Results

### 3.1. Effect of Grassland Phenological Stages on Ruminal Ammonia and Volatile Fatty Acid Concentrations

The values of the ruminal ammonia concentrations and the volatile fatty acids evaluated are presented in Table 1.

The highest concentration of RA was observed during the vegetative period, with 44.13 mg/dL, followed by the reproductive and dry periods, with a reduction of the order of 30% and 71%, respectively (*p* ≤ 0.05). The total concentration of VFAs averaged 138.82 mM, with no significant differences between phenological periods (*p* = 0.4133). Likewise, the acetate concentration did not differ between grassland phenological stages (*p* = 0.6223). However, the ruminal concentration of the rest of the VFAs analyzed was always higher in the vegetative period (*p* ≤ 0.05), highlighting the branched-chain isobutyric and isovaleric fatty acids, which presented significant differences between phenological periods, decreasing as the grassland passed from the vegetative to dry stage. Regarding the A:P ratio, this was 20% higher during the dry period compared to the average observed during the vegetative and reproductive periods (*p* ≤ 0.05) (Table 1).

### 3.2. Correlations between the Content of Fecal 2,6 Diaminopimelic Acid (DAPAf), Nitrogen (FN), Phosphorus (FP), and the Concentration of Volatile Fatty Acids (VFAs) and Ruminal Ammonia (RA)

All variables analyzed were normally distributed, so the degree of association between them was determined using Pearson’s correlation coefficient. The correlation coefficients between fecal indices, ruminal VFAs, and RA are presented in Table 2.

The RA was positively correlated with all VFAs evaluated individually (*p* ≤ 0.05), except for acetate. However, it did not have a significant correlation with the total VFAs, while the correlation with the A:P ratio was significant and negative.

The DAPAf concentration did not correlate with the concentration of the main ruminal volatile fatty acids or with their total VFAs (*p* > 0.05). However, it did present a positive correlation with the branched-chain fatty acids, isobutyric and isovaleric, which in both cases was of the order of 0.6 (*p* ≤ 0.05). A similar result was observed when establishing the correlation between DAPAf and RA concentration, whose coefficient was positive and significant (*p* ≤ 0.05). In the case of the correlation between DAPAf and the A:P ratio, it was significant and negative, with a coefficient of −0.73 (Table 2).

The FN concentration presented a positive and significant correlation with propionic acid (*p* ≤ 0.05) and very significant (*p* ≤ 0.01) with valeric, isobutyric, isovaleric acids, and with RA. In the case of the correlation between FN and the A:P ratio, it was significant and negative, with a coefficient of −0.74 (Table 2). However, this fecal indicator did not show a significant correlation (*p* > 0.05) with acetate, butyrate, and with total VFAs (Table 2).

The PF concentration presented a significant correlation (*p* ≤ 0.05) with all ruminal variables evaluated except for the concentration of acetic acid, propionic acid, and the total VFAs (Table 2). This fecal indicator showed a positive correlation with the concentration of isobutyric, butyric, isovaleric, and valeric acids, with coefficients greater than 0.7. The A:P relationship presented a negative correlation with the PF (*p* ≤ 0.05), while, with the RA concentration, this correlation was high and positive (*p* ≤ 0.01) (Table 2).

Table 3 shows the regression equations established between the fecal indicators and the VFAs and RA, which turned out to be significant (*p* ≤ 0.05).

Propionic acid was estimated by linear equations where only the FN and FP were the predictor variables, although with low R^2^ values. Butyric acid presented a significant regression equation only when FP was the predictor variable, with an adjustment close to 49%. In the case of valeric acid, both FN and FP were good predictors, obtaining significant linear regression equations, especially with FP, where R^2^ was close to 76% (Figure 3).

Unlike what happened with the previous variables, the linear regression equations that relate the concentrations of ruminal volatile branched-chain fatty acids and ruminal ammonia with fecal indicators, observed a high coefficient of determination (R^2^ > 80%) (Figure 3). The concentration of branched-chain VFAs, isobutyric and isovaleric, could be estimated by linear regression equations in which the three fecal indicators, especially the NF and PF, were used as predictive variables, obtaining a degree of adjustment greater than 87% (*p* ≤ 0.01) (Figure 3). A similar situation was observed in the RA concentration, where the NF and the PF presented regression equations with an R^2^ greater than 80% (*p* ≤ 0.01) (Figure 3).

The A:P ratio was estimated using linear equations, using the three fecal indices, although in this case the determination coefficients were of the order of 50% (Table 3).

## 4. Discussion

### 4.1. Effect of Grassland Phenological Stages on Ruminal Ammonia and Volatile Fatty Acid Concentrations

In all phenological stages evaluated, the RA concentration was above 5–6 mg/dL, considered optimal for both the production of microbial protein under in vitro conditions [27,28,29] as well as in vivo (3.5–25 mg/dL) [30]. Regarding other studies where the RA concentration was measured as a function of the changes experienced by the grassland with respect to the nutrient content, the minimum ruminal ammonia concentration during the dry period was similar to that found by Chun-tao et al. [31], who worked with lambs fed diets containing 14.7% of CP, as well as what was reported by Askar et al. [32] who mention ammonia concentrations of 12.1 mg/dL in the ruminal fluid of sheep feeding at natural grasslands during the dry season in the Ras Hederba Valley, Egypt. Likewise, the observed values were even higher than those found by Tatman et al. [33] in Rambouillet ewes fed ad libitum with wheat hay (CP = 9.7%; NDF = 65.3%; ADF = 45.3%) during the dry period of the grassland. The foregoing would allow us to affirm that the natural grassland in which we carried out our study could constitute a quality forage resource, even during the critical summer period.

The RA concentrations observed during the vegetative and reproductive periods of the grassland were higher than those indicated in the literature for sheep under grazing conditions. In this regard, Askar et al. [32] mention concentrations of the order of 24.6 mg/dL during the wet season of the grassland, while Adjorlolo et al. [34] indicate values close to 11 mg/dL for the rainy season in the Savannah Coastal Belt of Ghana in West Africa. In both studies aforementioned, these seasons correspond to the vegetative period. The high RA content observed during all grassland phenological stages, especially the vegetative and reproductive ones, would be closely associated with the high protein intake because of the CP concentration of the plant species that made up the grassland in those periods [16]. Plant grasses in semi-arid ecosystems vary widely in quantity and quality with subsequent effects on diet composition and selection [35]. The onset of maturity in the species that make up the grassland decreases digestibility and intake, reducing the CP and increasing the fiber intake [36]. The reduction in RA content recorded in this study as a function of the grassland’s phenological states was also observed by Estrada et al. [37], in steers consuming a semi-arid grassland in northern Mexico. The decrease in the CP content in the forage resource linked to the dry period reduces the amount of degradable protein, which results in a lower release of ammonia at the ruminal level, which is used by the microorganisms of the rumen for production of microbial protein [32,38].

The VFAs concentration in the rumen is determined by a balance between the production rate, the absorption rate through the ruminal wall, as well as by the rate of uptake of VFAs by ruminal microorganisms [39]. The acetic acid and total VFAs concentration remained stable during the three grassland phenological stages, being similar to those found by Askar et al. [32] in sheep grazing on natural grassland during the wet season (Ras Hederba Valley, Egypt), close to 140 mM, but higher than those reported by Tatman et al. [33] in sheep fed ad libitum with wheat straw (66.5 mM) and those found by Estrada et al. [37], which on average were of the order of 70 mM in steers grazing on a semi-arid grassland. The reduction in the propionic acid content in the dry period was also reported by Estrada et al. [37]. These VFAs are associated with the fermentation of soluble carbohydrates, which decrease as the grassland plants mature [32]. The same occurred in the case of butyric and valeric acids, whose concentration in the ruminal fluid most likely decreased because of the reduction in the availability of glucose for ruminal microorganisms in the grassland’s dry period. Related to the above, it should be mentioned that the formation of butyric acid in the rumen comes from the degradation of glucose towards acetoacetyl coA [18], so that an increase in the glucose available at the ruminal level would enhance the concentration of this fatty acid.

The reduction in the concentration of the branched-chain VFAs, isobutyric and isovaleric, would be directly associated with the amount of degradable protein at the ruminal level in each grassland phenological period [32,40], since they reflect the degradation of amino acids such as valine and leucine, respectively [41]. Regarding the protein degradation and the generation of branched-chain VFAs, Watson and Norton [42] found higher concentrations of isovaleric acid and ammonia in ruminal fluid of goats compared to sheep, subjected to the same diet. This was attributed to the greater proteolytic capacity of the microorganisms present in the goat’s rumen. The importance of branched-chain VFAs, isobutyric and isovaleric, is that they are used for the synthesis of branched-chain amino acids, being essential for the growth of various species of ruminal bacteria. Furthermore, these VFAs stimulate bacterial protein synthesis, increasing N retention in microbial bodies [41,43]. The increase in the A:P ratio as the grassland matured would be associated with the molar ratio of each of the VFAs produced in the rumen. In general, when the levels of cellulose and hemicellulose in the diet increase with respect to the levels of soluble carbohydrates and starch, the A:P ratio also increases [17]. In this case, the increase in cellulose and hemicellulose would be an effect of the phenological state on the diet selected by the animals.

### 4.2. Linear Correlations and Regressions between the Content of Fecal 2,6 Diaminopimelic Acid (DAPAf), Nitrogen (FN), Phosphorus (FP), and the Concentration of Volatile Fatty Acids (VFAs) and Ruminal Ammonia (RA)

The positive linear correlation found between RA and propionic, butyric, valeric, isobutyric, and isovaleric VFAs would be associated with the process of protein degradation at the ruminal level [18]. The RA concentration results from the degradation of degradable dietary protein and dietary non-protein nitrogen, from the hydrolysis of recycled urea into the rumen, and from the degradation of microbial crude protein [17]. In addition to ammonia, the degradation of amino acids forms volatile ruminal fatty acids, especially those with the branched-chain isobutyric and isovaleric acids, which is why high correlations are observed between these compounds. Butyric and valeric VFAs are also generated in this process, but in smaller quantities [18]. The positive linear correlation between RA and propionic acid could be the product of the degradation of the carbon chains present in the protein, as well as the higher consumption of carbohydrates associated with the high intake of degradable protein that occurs in grazing animals in these grasslands. In both cases, the metabolic pathway involved results in propionic acid, which is ultimately absorbed through the ruminal wall [18]. This would also explain the negative association found between RA and the A:P relationship; an increase in the concentration of AR would be associated with an increase in propionic acid, which decreases the A:P ratio.

Positive associations were observed between fecal indicators and certain VFAs and with the concentration of RA, so that the indicators could also reflect what happens at the ruminal level. A higher quantity and quality of the diet consumed has direct effects on the proliferation of ruminal microorganisms and on their fermentative capacity [18]. An increase in food intake, as well as metabolizable energy and crude protein, would increase the availability of substrates necessary for fermentative activity, especially for the predominant fibrolytic bacteria under the feeding conditions of the present study [44]. This increase in fermentation capacity is accompanied by an increase in microbial protein, part of which could be identified in the feces through NF [45], and DAPAf, in the case of the group of Gram-negative bacteria [46]. This greater fermentative activity is reflected in the positive association between fecal indicators and the concentration of certain VFAs, especially those with the branched-chain isobutyric and isovaleric acids, and RA. Bacteria require nitrogen, carbon chains, and energy to grow [18]. In this regard, Church [17] states that the amount of microbial protein that can be synthesized is limited by the amount of available energy to microbes and by the efficiency with which they use the available energy. The supply of protein and with it, of certain peptides and amino acids, serve as sources for the synthesis of branched-chain VFAs that are growth factors for cellulolytic bacteria [17], a situation that would explain the fact that all fecal indicators show significant correlations with isobutyric and isovaleric acids. Under in vitro conditions, the need for the presence of branched-chain fatty acids in the growth medium has been observed for the bacterial strain *R. flavefaciens* [47]. It has also been reported that the cellulolytic group of *Ruminococcus* has specific requirements for branched-chain VFAs [48,49].

In addition to the energy, protein, and the aforementioned VFAs, the availability of P is essential for bacterial growth, particularly those of the fibrolytic type [12,50,51], which is why the concentration of FP would also be a reflection of the fermentative capacity and would be directly associated with the variables linked to dry matter intake [16] and ruminal activity through the correlation indices found. In this regard, Kinclaid et al. [15] indicate that decreases in the supply of P in the rumen reduces the synthesis of microbial protein and thus the availability of amino acids for the animal. The FP was also associated in an important way to the intake of P, revealing a positive linear relationship between the excretion of P and the P consumed by the animals [16]. This situation was also observed in Suffolk Down sheep consuming diets with different P contents, where they found that the amount of FP corresponded to 66% of the P consumed, further confirming that in ruminants the route of fecal elimination of P is the most important [11].

Another of the ruminal fermentation products that showed high correlation with the three fecal indices was the RA. In addition to amino acids and branched-chain fatty acids, one of the important products derived from the proteolysis process is the RA [52]. The degradation of dietary protein in grazing ruminants occurs mainly by the action of proteolytic bacteria [17,18]. The increase in the RA product of proteolysis is associated with an increase in bacterial mass, which is why a higher content of FN and DAPAf is observed, as well as a higher concentration of isovaleric and isobutyric acids in the rumen. It is difficult to analyze each indicator separately, given the complexity of the ruminal system [18] and the different interactions that occur between the microorganisms present there. An example of this would be the high correlation found between FP and crude protein intake, as well as the degree of fit of the regression equation between both variables and the estimated association between FP and dry matter intake [16]. In this regard, it has been shown that diets deficient in P could lead to a decrease in voluntary food intake [53], which would lead to a modification in energy and protein intake that participate in different processes within the rumen. The importance of promoting the ruminal fermentation of organic matter and the synthesis of microbial biomass is that this fraction has the potential to satisfy between 70 and 85% of the energy requirements and between 70 and 100% of the protein requirements in ruminant [45].

Finally, it is important to highlight that the degree of fit obtained with the linear regression models that estimate the production of valeric, isobutyric, and isovaleric VFAs based on fecal N or P, were of a similar magnitude to that obtained with other techniques [54] and may be, therefore, complementary to these.

## 5. Conclusions

The fecal concentrations of nitrogen and phosphorus of sheep under extensive grazing conditions on Mediterranean annual grasslands are highly and positively correlated with the concentrations of the branched-chain volatile fatty acids and with that of ruminal ammonia. Therefore, these fecal indices could be used as predictors of ruminal activity and bacterial growth with a high degree of safety, using simple linear equations.

## Figures and Tables

**Figure 1 animals-12-00974-f001:**
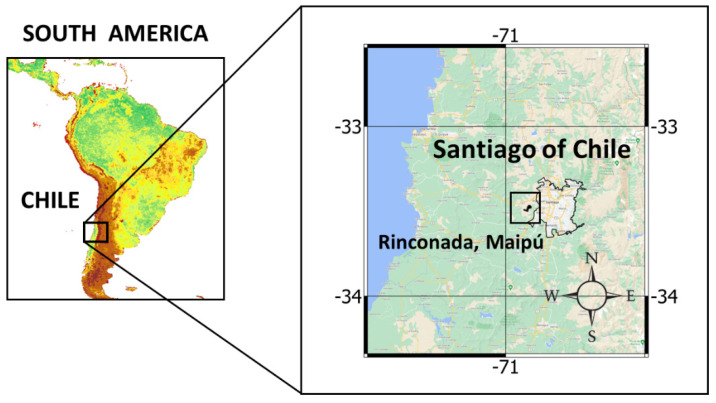
Location of the area where the study was carried out.

**Figure 2 animals-12-00974-f002:**
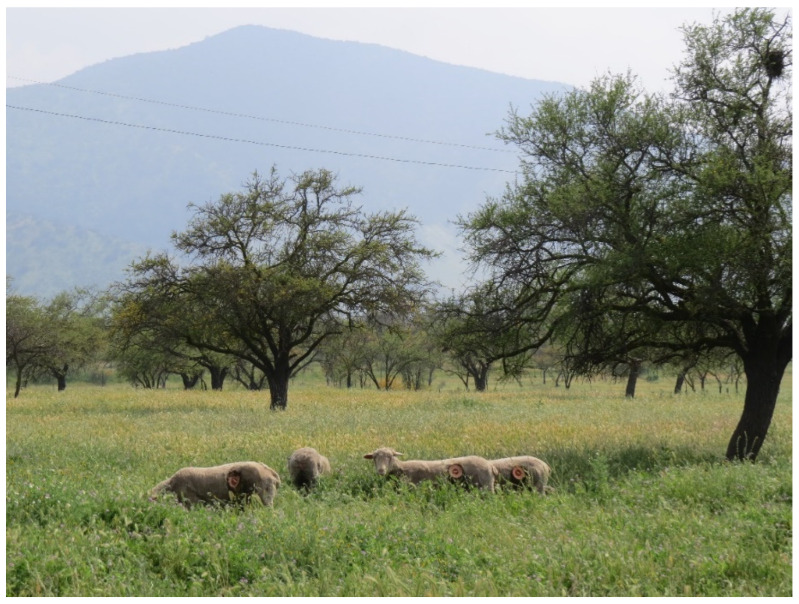
Sheep provided with a ruminal cannula, grazing on Mediterranean annual rangeland where the experimental study was carried out.

**Figure 3 animals-12-00974-f003:**
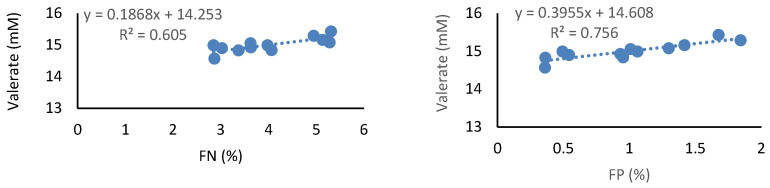
Significant linear regression equations with R^2^ > 60%, that estimate the concentration of ruminal valerate (mM), ruminal volatile branched-chain fatty acids (mM), and ruminal ammonia concentration (mg/dL), as a function of fecal nitrogen (FN, %) or phosphorus (FP, %), in extensive grazing sheep.

**Table 1 animals-12-00974-t001:** Concentration of ammonia (RA, mean ± standard deviation) and volatile fatty acids (VFAs, mean ± standard deviation) in grazing sheep ruminal fluid in each grassland phenological stage at the experimental site.

Ruminal Parameter	Grassland Phenological Stage		
	Vegetative	Reproductive	Dry
RA (mg/dL)	44.13 ± 2.98 ^a^	30.91 ± 1.27 ^b^	12.58 ± 8.42 ^c^
VFAs (mM)			
Acetate	43.67 ± 3.13 ^a^	43.34 ± 3.73 ^a^	46.19 ± 5.94 ^a^
Propionate	30.12 ± 2.09 ^a^	29.67 ± 3.00 ^a^	25.42 ± 1.55 ^b^
Butyrate	23.95 ± 0.86 ^a^	23.47 ± 0.81 ^ab^	20.77 ± 2.63 ^b^
Valerate	15.24 ± 0.15 ^a^	14.95 ± 0.09 ^b^	14.82 ± 0.18 ^b^
Isobutyrate	14.81 ± 0.08 ^a^	14.35 ± 0.09 ^b^	14.16 ± 0.09 ^c^
Isovalerate	14.36 ± 0.12 ^a^	13.74 ± 0.14 ^b^	13.42 ± 0.14 ^c^
Total VFAs	142.16 ± 4.12 ^a^	139.52 ± 7.11 ^a^	134.77 ± 10.22 ^a^
Acetate:Propionate ratio (A:P)	1.45 ± 0.05 ^b^	1.46 ± 0.08 ^b^	1.81 ± 0.13 ^a^

^a,b,c^ Different superscript letters indicate a statistically significant difference among grassland phenological periods, according to least significant difference (LSD) test (*p* ≤ 0.05).

**Table 2 animals-12-00974-t002:** Pearson’s correlation coefficients among the fecal contents of 2,6 diaminopimelic acid (DAPAf, mg/g MO), nitrogen (FN, %) and phosphorus (FP, %), and acetic (Ac), propionic (Prop), butyric (But), valeric (Val), isobutyric (Isobut), and isovaleric acids (Isoval), acetic:propionic ratio (A:P), total volatile fatty acids (Total VFAs, mM), and ruminal ammonia (RA, mg/dL) in extensive grazing sheep.

	Ac	Prop	But	Val	Isobut	Isoval	A:P	Total VFAs	RA
RA	−0.2341 ns	0.6732 *	0.6326 *	0.7784 **	0.8968 **	0.9202 **	−0.7909 **	0.4252 ns	---
	(*n* = 12)	(*n* = 12)	(*n* = 12)	(*n* = 12)	(*n* = 12)	(*n* = 12)	(*n* = 12)	(*n* = 12)	---
DAPAf	−0.4884 ns	0.3668 ns	0.2510 ns	0.5193 ns	0.5969 *	0.6359 *	−0.7274 *	0.0188ns	0.6724 *
	(*n* = 11)	(*n* = 11)	(*n* = 11)	(*n* = 11)	(*n* = 11)	(*n* = 11)	(*n* = 11)	(*n* = 11)	(*n* = 11)
FN	−0.1928 ns	0.6552 *	0.5350 ns	0.7780 **	0.9444 **	0.9438 **	−0.7379 **	0.4172 ns	0.9375 **
	(*n* = 12)	(*n* = 12)	(*n* = 12)	(*n* = 12)	(*n* = 12)	(*n* = 12)	(*n* = 12)	(*n* = 12)	(*n* = 12)
FP	−0.2693 ns	0.5623 ns	0.6992 *	0.8692 **	0.9374 **	0.9448 **	−0.7238 **	0.3852ns	0.9047 **
	(*n* = 12)	(*n* = 12)	(*n* = 12)	(*n* = 12)	(*n* = 12)	(*n* = 12)	(*n* = 12)	(*n* = 12)	(*n* = 12)

* *p* ≤ 0.05; ** *p* ≤ 0.01; ns = not significant.

**Table 3 animals-12-00974-t003:** Linear regression equations that estimate the concentration of ruminal volatile fatty acids (mM) and the content of ruminal ammonia (mg/dL) as a function of the concentration of the fecal indicators, 2,6 diaminopimelic acid (DAPAf, mg/g MO), fecal nitrogen (FN, %), and phosphorus (FP, %), in extensive grazing sheep. Coefficient of determination (R^2^), standard error of prediction (SEP), number of data (*n*), and *p*-value are included in the table.

Ruminal Parameter	Equation	R^2^	SEP	*n*	*p*-Value
Propionate	19.963 + 2.1061·FN	42.9	2.402	12	0.0208
	24.992 + 3.4258·FP	31.6	2.629	12	0.0570
Butyrate	19.797 + 2.9462·FP	48.9	1.572	12	0.0114
Valerate	14.253 + 0.1868 ·FN	60.5	0.149	12	0.0029
	14.608 + 0.3955·PF	75.6	0.117	12	0.0002
Isobutyrate	13.544 + 1.1516·DAPAf	35.6	0.261	11	0.0525
	13.246 + 0.2978·FN	89.2	0.103	12	≤0.0001
	13.882 + 0.5602·FP	87.9	0.109	12	≤0.0001
Isovalerate	12.464 + 1.7620·DAPAf	40.4	0.361	11	0.0355
	12.132 + 0.4261·FN	89.1	0.148	12	≤0.0001
	13.035 + 0.8083·FP	89.3	0.146	12	≤0.0001
A:P ratio	2.292 − 0.8916·DAPAf	52.9	0.142	11	0.0112
	2.182 − 0.1513·FN	54.4	0.137	12	0.0062
	1.856 − 0.2819·FP	52.4	0.140	12	0.0078
Ruminal ammonia	−20.466 + 62.9782·DAPAf	45.2	11.710	11	0.0234
	−27.814 + 14.2258·FN	87.9	5.223	12	≤0.0001
	3.285 + 26.0188·FP	81.9	6.392	12	0.0001

## Data Availability

The data on which these results are based can be requested directly from the authors.

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
