# Peer review of "Use of Fecal Indices as a Non-Invasive Tool for Ruminal Activity Evaluation in Extensive Grazing Sheep"

_animals, 2022, doi:10.3390/ani12080974_

Round 1

Reviewer 1 Report

Introduction

The introduction is badly structured. Please divide into paragraphs to make reading more reader-friendly.

Also, please summarize your hypothesis at the end. Also, please describe clearly in a separate paragraph the objectives of the work.

General comment. Please change the term Mediterranean. Obviously, it is wrong: Santiago to Barcelona: 11,500 km, Santiago to Beirut 13,500 km, hence please modify the term.

M & M

2.1. Redundant sub-section, but it can stay if the authors so wish.

2.2.

Without pregnancy or lactation: bad english

Feces sampling: same

2.4

Stool: used for humans only

2.5.

Please described which variables were found to have normal distribution and which did not.

Results

Please provide more graphs with results. The article is difficult to read and therefore figures will help readers to understand it.

Discussion

The section is badly written. Long paragraphs, incorrect English, wrong division of subsections.

Also, some recent relevant references are missing, please include.

In all, the manuscript needs a serious rewriting. Linguistic improvement must be a priority. Use of reader-friendly presentation will help.

Major revision and re-evaluation from the beginning.

Author Response

REVIEWER 1 RESPONSES

The introduction is badly structured. Please divide into paragraphs to make reading more reader-friendly.

Also, please summarize your hypothesis at the end. Also, please describe clearly in a separate paragraph the objectives of the work.

R:

The introduction is divided into three separate paragraphs, the last one expressing our hypothesis and objectives of the study.

General comment. Please change the term Mediterranean. Obviously, it is wrong: Santiago to Barcelona: 11,500 km, Santiago to Beirut 13,500 km, hence please modify the term.

R:

The term "Mediterranean" used in our study refers to the climatic type, which is characterized by cold and rainy winters and dry and hot summers. It is changed in the text by Mediterranean climate type.

2.1. Redundant sub-section, but it can stay if the authors so wish.

R:

The wording is maintained considering the sub-sections.

2.2.

Without pregnancy or lactation: bad English

R:

Changed to Dry ewes (non-pregnant and non-lactating)

Feces sampling: same

R:

Is changed to Fresh faeces samples.

2.4

Stool: used for humans only

R:

Term Stool is changed for faeces

2.5.

Please described which variables were found to have normal distribution and which did not.

R:

All variables analyzed were normally distributed, so the degree of association between them was determined using the Pearson’s correlation coefficient.

Results

Please provide more graphs with results. The article is difficult to read and therefore figures will help readers to understand it.

R:

A figure is added that accounts for the relationships between the fecal indicators and the ruminal variables that obtained high R2 (>80%)

Discussion

The section is badly written. Long paragraphs, incorrect English, wrong division of subsections.

Also, some recent relevant references are missing, please include.

R:

In the Discussion we consider three sections, since our results were presented in three sections:

4.1. Faecal Nitrogen (FN), Phosphorus (FP) and 2,6-Diaminomipelic Acid Concentrations (DAPAf) in the grassland

4.2. Effect of grassland phenological stages on ruminal ammonia and volatile fatty acids concentrations.

4.3. Linear correlations regressions between the content of faecal 2,6 diaminopimelic acid (DAPA), nitrogen (FN), phosphorus (FP) and the concentration of volatile fatty acids (VFAs) and ruminal ammonia (RA)

The paragraphs of each of these sections have been separated to facilitate reading.

Finally, a recent reference is included that shows that the linear equations obtained in our results, for the estimation of ruminal variables, have a similar degree of explanation than other analytical techniques.

In all, the manuscript needs a serious rewriting. Linguistic improvement must be a priority. Use of reader-friendly presentation will help.

Major revision and re-evaluation from the beginning.

R:

The manuscript will be sent for review in the English language via the system provided by MDPI.

Reviewer 2 Report

Review manuscript. Use of Fecal Indices as a Non-Invasive Tool for ruminal activity 2
Evaluation in Extensive-Grazing Sheep

The work deals with an interesting aspect

The manuscript may be published after taking into account the comments addressed to the
authors of the work Comments

1. An abstract written correctly contains the most important information

2. Introduction written correctly, the aim should be more clearly described (requires
editing)

3. Materials and methods: Please enter the body weight of the sheep, how old were the
animals. How long have they been in the pasture. Please describe sheep nutrition
(feeding system) in more detail. What was the pasture fleece (botanical composition)

4. The summary requires editing and reference to the results obtained.

The work can be published after minor revision.

Author Response

REVIEWER 2 RESPONSES

Review manuscript. Use of Fecal Indices as a Non-Invasive Tool for ruminal activity Evaluation in Extensive-Grazing Sheep

The work deals with an interesting aspect

The manuscript may be published after taking into account the comments addressed to the authors of the work Comments

  1. An abstract written correctly contains the most important information

R:

The summary is completed with relevant aspects of results and conclusions, indicating the magnitude of the correlations and their degree of significance.

  1. Introduction written correctly; the aim should be more clearly described (requires

editing)

R:

The introduction is separated into three paragraphs for a better understanding of the problem. The last paragraph explicitly defines the research hypothesis and objectives.

  1. Materials and methods: Please enter the body weight of the sheep, how old were the

animals. How long have they been in the pasture? Please describe sheep nutrition

(Feeding system) in more detail. What was the pasture fleece (botanical composition)

R:

The liveweight, age and physiological state of the sheep used in the study are added. It is indicated that the sheep continuously grazed on the pasture throughout the year, without receiving any type of supplementation.

  1. The summary requires editing and reference to the results obtained.

R:

The summary is completed with relevant aspects of results and conclusions, indicating the magnitude of the correlations and their degree of significance.

The work can be published after minor revision.

Round 2

Reviewer 1 Report

The manuscript has been significantly, but it still needs better restructuring, as well as linguistic improvements before final acceptance.

Author Response

dear reviewer:

We have revised the manuscript in response to your suggestions, for which we try to improve the writing and  structure of  the different sections (introduction, materials and methods, results, discussion and conclusions),

We also reviewed in detail the writing of the English, as well as the new numbering of the references.

All changes made are highlighted in the new draft

best regards
